# Diversity and Functional Distribution Characteristics of Myxobacterial Communities in the Rhizosphere of *Tamarix chinensis* Lour in Ebinur Lake Wetland, China

**DOI:** 10.3390/microorganisms11081924

**Published:** 2023-07-28

**Authors:** Xuemei Chen, Bo He, Cheng Ding, Xiaoyun Qi, Yang Li, Wenge Hu

**Affiliations:** School of Life Science, Shihezi University, Shihezi 832000, China; 18893705524@163.com (X.C.); hbjkml@163.com (B.H.); dc17699535485@163.com (C.D.); 15101294290@163.com (X.Q.); yangli33@tju.edu.cn (Y.L.)

**Keywords:** myxobacterial communities, Ebinur Lake Wetland, *Tamarix chinensis* Lour, diversity and functional

## Abstract

Soil salinity and desertification are seriously threatening the ecological environment of Ebinur Lake Wetland. Myxobacteria are the main soil microbes in this wetland. However, it is still unclear if the myxobacterial community structure and diversity can improve the ecological environment of Ebinur Lake Wetland by regulating soil nutrient cycling. Therefore, based on high-throughput sequencing of 16SrRNA gene technology, the composition, function, and diversity of the myxobacterial community in the rhizosphere of *Tamarix chinensis* Lour in Ebinur Lake Wetland were studied. Rhizosphere soil samples were collected from 10 sampling sites (S1, S2, S3, S4, S5, S6, S7, S8, S9, and S10) for three months (April, July, and October) to explore the main biotic and abiotic factors affecting the diversity and functions of myxobacterial communities. The results revealed that diversity of myxobacterial communities was mainly influenced by the seasons. The diversity of myxobacterial communities was significantly higher in the month of July, as compared to April and October. FAPROTAX functional prediction revealed that, in addition to predation or parasitic functions, myxobacteria were mainly involved in ecological functions, such as nitrite respiration, nitrite ammonification, and nitrogen respiration. The Spearman correlation analysis of the diversity and function of myxobacteria and bacteria showed that there were significant positive correlations between myxobacteria diversity, function, and bacterial diversity. The co-occurrence analysis of myxobacteria and bacterial networks showed that over time, myxobacteria interacted differently with different bacterial networks and jointly regulated the microbial community in the rhizosphere of *Tamarix chinensis* Lour through predation or cooperation. The redundancy analysis of soil physicochemical factors as well as the myxobacterial community showed that electrical conductivity, exchangeable calcium, and exchangeable potassium were the most important abiotic factors affecting the diversity, structure, and function of the myxobacterial community. These results reveal that myxobacteria may play important roles in degrading nitrogen compounds and regulating the activity of soil microorganisms. This study provides theoretical support for the ecological restoration of Ebinur Lake Wetland and lays the foundation for the future development and utilization of myxobacteria resources.

## 1. Introduction

Soil microbial community is an important part of the soil ecosystem, participating in the nutrient and energy flow in the soil ecosystem. The metabolic activities of soil microorganisms intensify soil respiration and increase the productivity of the soil ecosystem [1,2], thus promoting plant growth and development. Therefore, the biological function and diversity index of microbial communities are important parameters for directly depicting the changes in the activity of microbial community and explaining the characteristics and environmental conditions of microbial ecosystems [3]. Microbial diversity and species richness are crucial for maintaining the stability of soil ecosystems, and these parameters reflect changes in regional ecological environment to some extent [2,4].

Myxobacteria is a key group in the soil microbial food web and accounts for 4.10% of the bacterial community [5,6]. Myxobacteria can prey on a variety of microorganisms and are important predators that potentially regulate the structure and function of the soil microbial community and affect the microbial community dynamics in the environment [7]. Previous studies have shown that myxobacteria can affect the structure of microbial communities in the environment. Dai et al. [8] analyzed the myxobacterial community in four different fecal samples through high-throughput sequencing and found that the diversity of myxobacterial community was affected by the diversity of bacteria in different fecal samples. Furthermore, the type of fecal samples was reported as the main factor affecting the myxobacterial community structure in different fecal samples. Wang et al. [9] studied the diversity of myxobacteria in Dinghushan by combining high-throughput sequencing and culture methods and found that the biological factors affecting the diversity of Dinghushan myxobacteria were mainly soil bacterial diversity, while the key abiotic factors were soil pH and the available potassium content. Ye et al. [10] employed a *Corallococcus* sp. *EGB* to modify the soil microbial community structure and reduced the abundance of *Fusarium oxysporum* f. sp. cucumerinum in the soil by predation. Wang et al. [11] used 16S rRNA amplification and found that long-term nitrogen fertilizer application can significantly reduce the abundance, copy number, and diversity index of predatory myxobacteria. Soil acidification caused by nitrogen fertilizer application was reported as the main driving factor affecting the abundance and copy number of soil myxobacteria. Wang et al. [12] found that the addition of exogenous iron compounds can affect the distribution of anaerobic myxobacterial communities in rice soil samples. Wang et al. [13] found that the pH and Shannon index of predatory bacteria in farmland soil were significantly and positively correlated with the community structure of myxobacteria. Dai et al. [7] showed that the structure and diversity of soil microbial communities were closely related to soil nutrient cycle. The diversity index, species richness, and nearest taxon index of soil microbial communities in farmland were significantly and negatively correlated with soil nutrient cycling.

The Ebinur Lake Wetland is located in the desert area of the Junggar Basin in the north of Xinjiang. It is a typical lake wetland in the arid inland area of China. In recent years, climate change and human activities have caused serious ecological problems, such as sharp decline in the ecological diversity of the Ebinur Lake Wetland, aggravation of the salinization in the lakeside wetland, and expansion of desertification [14,15]. *Tamarix chinensis* is a commonly found desert plant in the region [16]. Its unique salt-secreting glands enables the leaf surface to discharge salt, thus regulating the ion balance in plants, maintaining the stability of osmotic pressure, improving the salt tolerance of plants, and increasing the concentration of soil nutrients [17].

Therefore, the present study aims to (1) study the changes in the structure and diversity of the rhizosphere myxobacterial community as well as the effects of soil physicochemical factors on *Tamarix* in order to reflect the changes in the ecological environment of the Ebinur Lake Wetland, (2) study the functional distribution of myxobacteria and the symbiotic relationship between myxobacteria and bacterial networks in order to understand the regulatory role of myxobacteria in wetland ecosystems, and (3) study the main biological and abiotic factors affecting the diversity and function of myxobacteria to understand the role of myxobacteria in soil nutrient cycling. This study provides theoretical support for improving wetland salinization, improving soil microbial functions, and promoting the restoration of ecological vegetation in Ebinur Lake Wetland. The study lays the foundation for the future development and utilization of the myxobacterial community.

## 2. Materials and Methods

### 2.1. Research Location

The rhizosphere soil samples of *Tamarix chinensis* Lour were collected from 10 sites located around the three main inflow rivers of the Ebinur Lake Wetland (Bortala Mongol Autonomous Prefecture, Jinghe River, and Kuitun River). These 10 sites (S1, S2, S3, S4, S5, S6, S7, S8, S9, and S10) were selected by the global positioning system (GPS), and the latitude and longitude of these sampling sites were recorded. Samples were collected from the sites in the month of April, July, and October. A five-point sampling method was adopted to collect the rhizosphere soil samples of *Tamarix chinensis* Lour from each site. Five sampling points were selected at each site, and three samples were collected from each point. 15 soil samples collected from each sampling site were mixed evenly. Soil samples were placed in a portable refrigerator and brought back to the laboratory. Part of each soil sample was stored at −80 °C for genomic DNA extraction, and the remaining part was naturally dried and stored at 4 °C for analysis of physicochemical properties of soil. The details of soil samples have been provided in Figure 1.

### 2.2. Soil Physicochemical Properties

A pH meter was used to measure the pH of soil in a 1:2.5 soil water solution. Moisture content (MC) of soil was determined by the drying method. Electrical conductivity (EC) was measured by using a DDSJ-308A conductivity meter in a 1:5 soil water solution. Organic matter (OM) in soil was determined through potassium dichromate volumetric method and heat of dilution method [7]. Available phosphorus (AP) was measured using the 0.5 mol/L NaHCO_3_ extraction molybdenum antimony colorimetric method, while available potassium (AK) was determined through the NaNO_3_ extraction sodium tetraphenylboron turbidimetry method. Inorganic nitrogen (IN) was measured by the 2 mol/L KCl extraction indophenol blue colorimetric method. Exchangeable calcium (CI) and magnesium (MI) in soil were determined by using ethylenediamine tetraacetic acid titration method [2,7,8,18].

### 2.3. Soil DNA Extraction

Soil total DNA was extracted by using an improved DNeasy^®^PoweeSoil^®^Pro kit [19]. After DNA isolation, the V4-V5 variable region of the 16SrRNA bacterial gene was amplified using primers: 515F (5′-GTGNCAGCMGCCGCGGTAA-3′) and 907R (5′-CCGTCAATTCMTTTRAGTTT-3′) [20]. The polymerase chain reaction (PCR) was carried out using 25 μL reaction mixture including 12.5 μL of 2× Taq Master Mix, 0.5 μL of each primer (0.2 μmol/L), 10 ng/L DNA sample, and 10.5 μL ddH_2_O. The PCR amplification cycles consisted of pre-denaturation (95 °C, 3 min), denaturation (94 °C, 30 s), annealing (55 °C, 30 s), extension (72 °C, 45 s), and finally extension (72 °C, 10 min), with a 4 °C hold. The entire amplification process consisted of 30 PCR cycles. The PCR products of three replicates of samples were combined, and the concentration and specificity of the amplified DNA fragments were detected by 1% agarose gel electrophoresis. After passing the quality test, PCR products were used for the subsequent high-throughput sequencing. The sequencing library was generated by using the TruseqDNA PCR-Free Sample Preparation Kit (Hayward, Illumina, San Diego, CA, USA), and the index code was added. The quality of the sequencing library was evaluated on a qubit@2.0 fluorometer (Thermoscientific, Waltham, MA, USA) and a Bioanalyzer 2100 (Palo Alto, Agilent Technologies, Santa Clara, CA, USA). The gene library was sequenced on the Illumina NovaSeq platform by IlluminaPE250 (Novogene, Beijing, China).

### 2.4. Data Processing and Analysis

Quality control was performed on the raw data obtained from the sequencing of the 16SrRNA gene amplicon. Operational taxonomic units (OTUs) were classified based on 97% similarity. Using MUSCLE software (Version 3.8.31) combined with the Silva132 database for species annotation (threshold 0.8–1), OTUs annotated as myxobacteria at each taxonomic level were selected. Based on existing literature regarding the classification of myxobacteria, species information and community composition of myxobacteria at each taxonomic level were obtained. Origin 2021 was used to draw the column stacking map of myxobacteria at the genus level. Based on the level of OTUs, QIIME (Version 1.9.1) was used to analyze the α and β diversity of myxobacteria in the rhizosphere soil of *Tamarix chinensis* Lour. The temporal and spatial distribution characteristics of myxobacteria diversity in the rhizosphere soil were characterized using the α diversity index (Shannon index, Simpson index, Chao1 index, ACE), while the principal component analysis of the genus-level myxobacterial community was carried out based on Bray–Curtis distance. The functions of myxobacteria in the rhizosphere of *Tamarix ramosissima* in Ebinur Lake Wetland were predicted using the functional annotation of the prokaryotic taxon (FAPROTAX 1.2.4). The main biological factors affecting the function and diversity of myxobacteria were analyzed using Spearman correlation heat maps. Based on Spearman correlation, WeKemo Biotech Cloud Network Service (https://bioincloud.tech/) (access on 5 May 2023) was used to calculate the parameters of the microbial network. A microbial co-occurrence network was constructed by Gephi0.10.1. to analyze the main myxobacteria genera that affect other bacteria. Canoco5.0 was used for redundancy analysis to identify the main abiotic factors that affect the diversity, function, and genus level community composition of myxobacteria. High-throughput sequencing data have been deposited in the National Center for Biotechnology Information Sequence Read Archive (BioProject ID PRJNA997573, Submission: SUB13700201).

## 3. Results

### 3.1. Temporal and Spatial Distribution of Myxobacterial Communities in the Rhizosphere of Tamarix chinensis Lour in Ebinur Lake Wetland

#### 3.1.1. Genus Level Composition of Myxobacteria Community

The sequence number of myxobacteria identified in the rhizosphere soil of *Tamarix chinensis* Lour fluctuated in the range of 91~1493 tags, accounting for 0.29–2.78% of the sequence number of all bacteria in the soil samples collected from various sampling points. Except for undefined No_Rank Myxo P3OB-42, 1 order, 3 suborders, 8 families, and 11 genera were identified from each sample. The three suborders were Cystobacterineae, Sorangiineae, and Nannocystineae. The eight families were Myxococcaceae, Sandaracinaceae, Haliangiaceae, Nannocystaceae, Anaeromyxobacteraceae, Polyangiaceae, Vulgatibacteraceae, and Phaselicystidaceae. The 11 genera were *Archangium*, *Haliangium*, *Anaeromyxobacter*, *Enhygromyxa*, *Nannocystis*, *Pajaroellobacter*, *Myxococcus*, *Vulgatibacter*, *Phaselicystis*, *Chondromyces,* and *Sandaracinus*. The spatial variations in the myxobacterial community at the genus level are shown in Figure 2A. The relative abundance of *Archangium* in all samples ranged from 11.72% to 36.63%, with the highest relative abundance in the sample collected from the S9 site and the lowest in the sample from the S4 site. The relative abundance of *Haliangium* ranged from 14.65% to 30.40%, with the highest and lowest relative abundances observed in the S10 and S6 samples, respectively. The relative abundance of *Sandaracinus* ranged from 2.56% to 5.86% (highest in the S4 sample and lowest in the S1 and S5 samples). The relative abundance of *Anaeromyxobacter* ranged from 0% to 4.03%, with the highest relative abundance in the S10 sample. The relative abundance of *Enhygromyxa* ranged from 1.10% to 4.76% (highest in the S8 sample and lowest in the S5 samples). The relative abundance of *Nannocystis* ranged from 1.10% to 4.40% (highest in the S2 sample and lowest in the S3 and S6 samples). Similarly, relative abundances of *Pajaroelobacter, Vulgatibacter,* and *Myxococcus* ranged from 0% to 1.83%, 0% to 1.83%, and 0.37% to 2.20%, respectively. The relative abundances of *Pajaroelobacter, Vulgatibacter,* and *Myxococcus* were high in the S3 sample.

The changes in the myxobacterial community structure in terms of the top 10 genera according to relative abundance are shown in Figure 2B. The relative abundance of *Archangium* gradually decreased over time, with the highest relative abundance observed in April (21.54%). On the contrary, the relative abundance of *Haliangium* gradually increased over time, with the highest relative abundance observed in October (28.13%). *Archangium* and *Haliangium* were the dominant myxobacterial genera during the three months of research. Furthermore, high relative abundance of *Sandaracinus*, *Anaeromyxobate*, *Enhygromyxa,* and *Nannocystis* genera were also observed in July and October.

#### 3.1.2. Diversity of Myxobacteria in the Rhizosphere of *Tamarix chinensis* Lour in Ebinur Lake Wetland

The spatial and temporal distributions of the α diversity index of the myxobacterial community in the rhizosphere of *Tamarix chinensis* Lour are shown in Figure 3 and Figure 4. The spatial distribution of the diversity index showed that the Shannon index, which reflects the diversity of the myxobacterial community, ranged from 4.18 to 5.02, while the Simpson index ranged from 0.85 to 0.95. The Shannon index and Simpson index were relatively high for the S2 and S8 samples, and lowest for the S9 sample. This indicated that the rhizosphere soil in S2 and S8 sites had the highest diversity of the myxobacterial community, while the S9 site had the lowest diversity of myxobacteria in rhizosphere soil. The spatial distribution of the richness index of the myxobacterial community showed that S10 samples had the highest Chao1 index (99.64) and ACE index (98.61), while S5 and S7 samples had the lowest Chao1 index (64.20 and 68.17, respectively) and ACE index (85.15 and 72.30, respectively). These observations indicated that the S10 site had the highest abundance of the myxobacterial community, while S5 and S7 sites had the lowest abundance of the myxobacterial community in the rhizosphere soil.

The time distribution of the diversity index showed that there was no significant difference in the Shannon index and Simpson index of the myxobacterial community in the three months (Figure 4). The Shannon index and Simpson index of the myxobacterial community were highest in July (4.949 and 0.91, respectively). Only small differences were observed in both indices during April and October. This indicated that the diversity of the myxobacterial community was highest in July. The temporal distribution of the richness of the myxobacterial community showed that the Chao1 index and ACE index were highest in April (85.051 and 94.373, respectively) and lowest in October (63.704 and 73.421, respectively). This indicated that the myxobacterial community richness was highest in April and lowest in October.

Based on Bray–Curtis distances, principal component analysis was conducted on the genus level myxobacterial community (Figure 5). The results showed that there were significant differences in the temporal distribution of myxobacteria communities in the rhizosphere of *Tamarix chinensis* Lour. (PERMANOVA: F = 2.453, *p* = 0.024; Appendix A). There was no significant difference between the spatial structure of the myxobacterial communities (PERMANOVA: F = 1.14, *p* = 0.318; Appendix A). Among all the samples, S7, S5, and S9 showed significant differences in the structure of the myxobacterial community, compared to other samples. The differences in the community structure of myxobacteria in other samples were relatively small. The community structure of myxobacteria was mainly influenced by seasonal changes.

#### 3.1.3. Functional Diversity of Myxobacteria in the Rhizosphere of Tamarix chinensis Lour in Ebinur Lake Wetland

The functional heatmap distribution of FAPROTAX in the rhizospheres of *Tamarix chinensis* Lour in Ebinur Lake Wetland is shown in Figure 6. The results showed that the myxobacteria of all soil samples participated in predatory or exoparasitic, chemoheterotrophic, and cell lysis functions in all three months. However, functional abundance of myxobacteria in the *Tamarix ramosissima* rhizosphere was significantly higher in October than in April and July for nitrogen cycle functions such as predation, parasitism, or nitrate respiration. The functional abundances of the myxobacterial community in S4, S5, and S9 samples were significantly higher than the other samples for nitrate respiration, fumarate respiration, iron respiration, and other nitrogen cycle functions. This indicates that myxobacteria in the rhizosphere of *Tamarix chinensis* Lour in Ebinur Lake Wetland have the potential to regulate the wetland ecological environment.

### 3.2. Association between the Diversity of Sticky Bacteria and Bacterial Diversity in the Rhizospheres of Tamarix chinensis Lour in Ebinur Lake Wetland

#### 3.2.1. Correlation between the Diversity Indexes of Myxobacteria and Bacteria

The Spearman correlation between the diversity indexes of myxobacteria and bacteria in the rhizospheres of *Tamarix chinensis* Lour in the Ebinur Lake Wetland is shown in Figure 7. It was found that the Shannon index of myxobacteria was significantly and positively correlated with the Shannon index of bacteria (*p* < 0.05). Similarly, the Chao1 index of myxobacteria was significantly positively correlated with the Chao1 index of bacteria. Significant positive correlation was also observed between the ACE indexes of myxobacteria and bacteria. These observations indicated that the diversity of myxobacteria was affected by the bacterial diversity. Therefore, myxobacteria diversity was determined as the main biological factor affecting the diversity of bacterial communities in *Tamarix chinensis* Lour rhizospheres in Ebinur Lake Wetland.

#### 3.2.2. Function and Bacterial Diversity of Myxobacteria of *Tamarix chinensis* Lour Rhizospheres in Ebinur Lake Wetland

The correlations between the functional abundance of myxobacteria and diversity index of bacteria and myxobacteria are shown in Figure 8. Except for the aerobic chemoheterotrophy, the functional abundance of myxobacteria showed a significant correlation with the diversity index of myxobacteria and bacteria. Chemoheterotrophy and cellulolysis functions were found to be negatively correlated with the Simpson index (*p* < 0.001) and Shannon index of myxobacteria (*p* < 0.05). Functions related to nitrogen cycle, such as nitrite respiration, nitrite ammonification, nitrogen respiration, and other functional abundances, were found to be significantly and negatively correlated with the bacterial diversity indices (i.e., Shannon, Simpson, Chao1, ACE). These observations indicated that bacterial diversity was the main factor affecting the nitrogen cycle function of myxobacteria.

#### 3.2.3. Interactions between Myxobacteria and Bacterial Community Network of *Tamarix chinensis* Lour Rhizospheres in Ebinur Lake Wetland

Based on Spearman correlation (correlation coefficients r ≥ 0.6 and r ≤ −0.6), the co-occurrence of dominant myxobacteria and bacterial networks in the rhizospheres of *Tamarix chinensis* Lour was analyzed in different seasons. Dominant myxobacteria and the top 20 bacteria in terms of relative abundance were selected (Figure 9).

In April, the microbial network consisted of 29 nodes and 75 sides (Table 1). *Lentilactobacillus* (43.07%) and *Shewanella* (41.66%), belonging to Firmicutes and Proteobacteria, respectively, were found in relatively high abundance. Different dominant myxobacteria interacted differently with dominant bacteria. *Haliangium* myxobacteria showed significant positive correlations (*p* < 0.05), with bacteria such as *Streptomyces* and *Bacillus*. On the contrary, significant negative correlations were found between *Haliangium* and dominant bacteria, such as *Pseudomonas, Acinetobacter*; Similarly, *Anaeromyxoactor*, *Sandaracinus*, and *Enhygromyxa* myxobacteria, which showed significant negative correlations with *Lentilactobacillus*.

In July, the microbial network was composed of 24 nodes and 32 sides. *Pseudomonas* (37.76%) and *Halomonas* (20.01%) belonging to the Proteobacteria phylum were found in relatively high abundance. *Myxococcus*, a secondary dominant myxobacterium, showed significant positive correlations with *Pseudomonas*, while *Nannocystis* myxobacteria showed significant negative correlation with *Halomonas*.

In October, the microbial network was composed of 25 nodes and 37 sides. *Pseudomonas* (67.23%) and *Alifidinibius* (20.38%), belonging to Proteobacteria and Bacteroidota phyla, respectively, were found in relatively high abundance. *Pajaroellobacter* and *Myxococcus* were found to be significantly and negatively correlated with *Alifidinibius*. These observations indicated a potential cooperative or predatory relationship between the dominant myxobacteria and dominant bacteria for jointly regulating the dynamics of wetland microbiota. By comparing the interactions between dominant myxobacteria and bacteria in three seasons, it was found that the interaction between *Haliangium* and dominant bacteria gradually weakened. However, the negative correlation between secondary dominant myxobacteria, such as *Myxococcus* and *Pajaroelobacter,* and other bacteria gradually strengthened. This indicates that the myxobacteria in the rhizosphere of *Tamarix* in wetlands mainly regulate the bacterial community through cooperation or predatory function.

### 3.3. Correlation between Rhizosphere Myxobacteria and Soil Physicochemical Factors of Tamarix chinensis Lour in Ebinur Lake Wetland

The redundancy analysis of soil physicochemical factors and diversity of myxobacteria in the rhizosphere of *Tamarix chinensis* Lour in the Ebinur Lake Wetland are shown in Figure 10. Canoco5.0 was used to screen the soil physicochemical factors that significantly affect the structure, diversity index, and function of the myxobacterial community in the rhizospheres. Redundancy analysis was conducted using the screened soil physicochemical factors (Appendix A). The results showed that soil physicochemical factors can affect the structure, diversity, and function of the myxobacterial community in the wetland. The two axes explained 14.33%, 14.43%, and 25.23% of the total variations in the structure, diversity, and function of soil bacterial communities, respectively. EC (*p* = 0.008, contribution = 54.5%) and MI (*p* = 0.052, contribution = 45.5%) explained 7.8% and 6.8% of the variations in the community structure of myxobacteria in the rhizosphere of *Tamarix chinensis* Lour, respectively. EC and MI significantly affected the distribution of myxobacteria, such as *Enhygromyxa, Pajaroelobacter, Archangium*, and *Myxococcus*, in the community. This indicated that EC was the main abiotic factor affecting the community structure of myxobacteria and regulating the distribution of myxobacteria in the rhizosphere of *Tamarix chinensis* Lour. CI (*p* = 0.066, contribution = 61.2%) and MC (*p* = 0.168, contribution = 38.2%) explained the 8.8% and 5.6% variations in the diversity of myxobacterial communities in *Tamarix* rhizospheres, respectively. This indicated that CI was the main abiotic factor affecting the diversity of myxobacteria in the rhizospheres of *Tamarix*. Similarly, AK (*p* = 0.046; contribution = 31.6%) and AP (*p* = 0.036; contribution = 24.3%) explained 14.3% and 11% variations in the functional diversity of myxobacterial communities, respectively. This suggested that AK was the main abiotic factor affecting the functional diversity of myxobacteria in the rhizosphere of *Tamarix chinensis* Lour. These functions included nitrite_respiration, nitrate_ammonification, nitrogen_respiration, nitrite_ammonification, and nitrate_respiration. Abundance of function such as recruitment was significantly positively correlated with AK (*p* < 0.05), indicating that AK was able to promote the functional abundance of myxobacteria related to nitrogen cycle and to regulate the expression of functional genes in myxobacteria.

## 4. Discussion

### 4.1. Community Structure and Functional Distribution of Myxobacteria in the Rhizosphere of Tamarix chinensis Lour in Ebinur Lake Wetland

This study used 16SrRNA gene amplicon sequencing technology to identify the sequence number of myxobacteria in rhizosphere soil samples of *Tamarix chinensis* Lour collected from different sites, which accounted for 0.29% to 2.78% of wetland soil bacteria. Wang et al. found that the sequence number of myxobacteria in different soil samples accounted for 0.92% to 2.24% of the total bacterial population [9], indicating that myxobacteria dominated in most soil samples. The Shannon index, which reflects the diversity in the myxobacterial community, varied between 4.18 and 5.02. Wang et al. [9] used high-throughput amplicon sequencing technology to find that the Shannon index of the myxobacterial community in the acidic soil of Dinghushan ranged from 1.95 to 3.15, indicating that the diversity of myxobacteria in alkaline soil was higher than the acidic soil. Such changes in the diversity and composition of the myxobacterial community definitely affect the interaction of microbiomes with nutrient cycling and plant growth in saline soils. The myxobacteria diversity showed significant positive correlation with bacterial diversity, as well as significant negative correlations with the exchangeable calcium ions in soil. Relevant studies found that a high concentration of calcium ions may promote the growth of some bacteria that exert antagonistic effects on myxobacteria, thus affecting the diversity of the myxobacterial community. The diversity and relevance of soil microbial communities are sensitive to environmental changes and are closely related to the nutrient cycling, as well as soil and ecosystem functions provided by microbial communities [21]. The relative abundance of *Haliangium*, a salt tolerant myxobacteria [22], gradually increased over time, indicating that the salinization of the *Tamarix* rhizosphere soil in the Ebinur Lake Wetland worsens over time. It has been found that the diversity of the myxobacterial community in the rhizosphere of *Tamarix ramosissima* in the Ebinur Lake Wetland was highest in July, which may be due to the high temperature in that month, affecting the microbial enzyme activity in the soil [23], accelerating the mineralization and nitrification of the soil [24], and providing sufficient nutrients for the growth of myxobacteria. The prediction of the FAPROTAX function of myxobacteria in the rhizosphere soil samples of *Tamarix chinensis* Lour revealed that the myxobacteria in all samples had the highest abundance of predatory or exoparasitic, chemoheterotrophic, and cellulolysis functions. The S6 soil sample showed the highest abundance of predation or parasitical functions in myxobacterial communities. Moreover, the predation or parasitical functions of myxobacteria in the S6 sample were related to soil biological health. This finding suggested that S6 soil sample myxobacteria were able to regulate soil microorganisms through predation. In October, the rhizosphere soil samples of *Tamarix chinensis* Lour showed a high abundance of functions related to nitrogen cycle (such as nitrate reduction, nitrogen respiration, and nitrate respiration). This indicated that the myxobacteria in rhizospheres of *Tamarix chinensis* Lour mainly participated in the removal of NO_2_^−^, NO_3_^−^, and NH_4_^+^ in October. This finding confirmed the potential role of some myxobacteria in having potential ecological roles in degrading nitrogen compounds and regulating soil microbial activity. *Sandaracinacae* is an autotrophic bacterium that can utilize ethanol, hydrogen, butyrate [8], and other low molecular energy. It has been found that *Anaeromyxobacter dehalogenans* 2CP-C can cause coupling of acetate [25] or carry out anaerobic respiration with organic halides, soluble and amorphous iron oxides, nitrates, nitrites, and fumaric acid to perform anaerobic respiration [26]. *Anaeromyxobacter* (an important N-fixing microorganism [27]) and *Haliangium* have potential roles in denitrification and P solubilization [7,28]. The co-occurrence of microbial networks revealed that *Anaeromyxoactor* and *Sandaracinus* myxobacteria showed significant negative correlations with *Lentilactobacillu*, indicating a potential predator–prey relationship between these bacteria. Thus, functional myxobacteria can regulate the homeostasis of microflora through the interaction with special bacteria.

### 4.2. Response of Myxobacteria Community to Biological Factors

In this study, α diversity indexes of myxobacteria and bacterial community were significantly and positively correlated, indicating that the diversity of the myxobacterial community was influenced by the diversity of the bacterial community. This may have been caused by the interaction between bacteria and myxobacteria. Myxobacteria are predatory microbes that fulfill their nutritional requirements by preying on other bacteria in the environment [8]. Predation is considered an important evolutionary and ecological force that can affect the microbial community structure and function of ecological systems [13,29]. Since soil myxobacteria cannot synthesize riboflavin and branched-chain amino acids on their own, they prey on the bacteria that are able to synthesize these substances [30]. Therefore, the microbial community structure and diversity may be influenced by the diversity of prey bacteria [29]. Furthermore, the predation preference of the myxobacteria has the potential to explain the effect of bacterial diversity on myxobacteria community structure [8,30]. We also observed that a negative correlation between the bacterial diversity was negatively correlated with the functional abundance of myxobacteria, such as nitrite respiration, nitrite ammonification, and nitrogen respiration. This observation was also in agreement with the findings [7,31]. The network symbiosis results showed that *Pseudomonas*, *Shewanella*, and *Halomonas* were the core microbial flora in the *Tamarix chinensis* Lour rhizosphere, belonging to the Proteobacteria phylum. *Pseudomonas*, *Shewanella*, and *Halomonas* interacted differently with myxobacteria, indicating the dynamic balance between Proteobacteria and myxobacteria through cooperation or predation. Dai et al. found that Myxococcota were correlated with Proteobacteria and Nitrospirota [7], which are potential drivers of soil multi-nutrient cycling in reforested ecosystems [31]. Microbes of the Pseudomonadaceae family can produce indole acetic acid and solubilize mineral phosphate, thereby promoting plant growth and participating in soil nutrient cycling [32].

### 4.3. Response of Myxobacterium Communities to Abiotic Factors

Soil physicochemical factors can directly or indirectly affect the structure and diversity of soil bacterial communities [33]. Based on redundancy analysis, it was found that CI and MC were the main factors affecting the myxobacterial community in rhizospheres of *Tamarix* in the Ebinur Lake Wetland. CI was the most important factor affecting the α diversity of the myxobacterial community. Soil exchangeable Ca^2+^ content was negatively correlated with the diversity index of myxobacterial communities, indicating that high concentrations of Ca^2+^can reduce the diversity of myxobacterial communities. This finding was consistent with the results reported by Dai et al. [8]. EC was the most important factor affecting the structure of myxobacteria in the rhizosphere of *Tamarix* in the wetland. Relevant studies [7] showed that pH, TK, SOC, and TP are the main drivers of myxobacterial community assembly. Dai et al. [8] found a significant negative correlation between NH_4_^+^- N concentration and the structure of the myxobacterial community, which was similar to the results obtained in this study. In addition to the content of exchangeable calcium and magnesium ions in the soil, certain metal ions in the soil may affect the activity of myxobacteria. Wang et al. [31] suggested that the differences in the inherent amorphous iron oxide content in paddy soil may also determine the unique community structure and succession of anaerobic bacteria in paddy soil. Furthermore, stochastic processes (such as immigration, mutations, and extinction) also contribute to variation in the bacterial community [34,35]. Abiotic factors, such as MC and OM, had no significant effect on the myxobacterial community. This can be attributed to the ability of myxobacteria to form myxospores, which show strong resistance during harsh environments and contribute to the extensive adaptability of myxobacteria. This may weaken the influence of environmental factors on myxobacteria [34]. Secondly, in this study, the rhizosphere soil samples were alkaline (pH 8.05~8.95), and OM in the soil was relatively low (3.46 g/kg~43.09 g/kg). Myxobacteria are generally found in the environment with a pH of 6.5~8.5. Some myxobacteria may not adapt to live in the alkaline soil, especially in neutral–weak alkaline soil with a pH of 6.0~8.0 [8,36]. In addition, myxobacteria are predatory microbes that fulfill their nutritional requirements by preying on other bacteria in the environment. Therefore, their nutrient intake from the environment may be relatively low. Moreover, the 515F and 907R primers used in this study are not universal primers for all myxobacteria, and their coverage needs to be improved [11]. Moreover, the interactions between host plants and soil microorganisms are complex [2]. Therefore, interactions of *Tamarix chinensis* Lour with myxobacteria and the surrounding environment needs further exploration. In addition to the content of CI and MI in the soil, certain metal ions in the soil may also affect the activity of myxobacteria.

## 5. Conclusions

The myxobacteria sequences identified from the rhizospheres of *Tamarix chinensis* Lour in the Ebinur Lake Wetland belonged to 1 order, 3 suborders, 9 families (90% of known families), and 11 genera. The community structure of myxobacteria at the genus level changed dynamically with time and space. *Archangium* and *Haliangium* were the dominant myxobacteria in all soil samples, and the relative abundance of *Haliangium* gradually increased with time. The nutrient content in the rhizosphere soil was relatively low, and the soil was saline alkaline in nature. Salinization gradually intensified with the changing seasons. The diversity and function of myxobacteria were mainly affected by the diversity of bacteria. The interaction between dominant myxobacteria and bacterial network varied in different seasons. Through predation or cooperation, myxobacteria and bacteria jointly regulated the dynamics of microbial community in the rhizospheres of *Tamarix chinensis* Lour. It has been found in this study that the microbial community function in rhizosphere soils is mainly related to the nitrogen cycle, and myxobacteria plays an important role in promoting the soil nitrogen cycle. EC, CI, and AK were determined as the main abiotic factors affecting the diversity, structure, and function of myxobacterial communities.

## Figures and Tables

**Figure 1 microorganisms-11-01924-f001:**
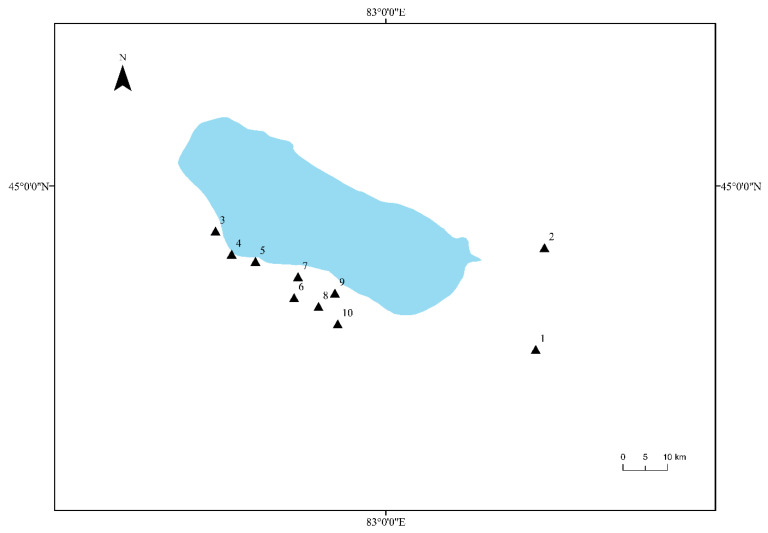
Distribution map of *Tamarix chinensis* Lour rhizosphere samples in the Ebinur Lake Wetland. The Arabic numbers 1–10 indicate the distribution locations of all sampling sites.

**Figure 2 microorganisms-11-01924-f002:**
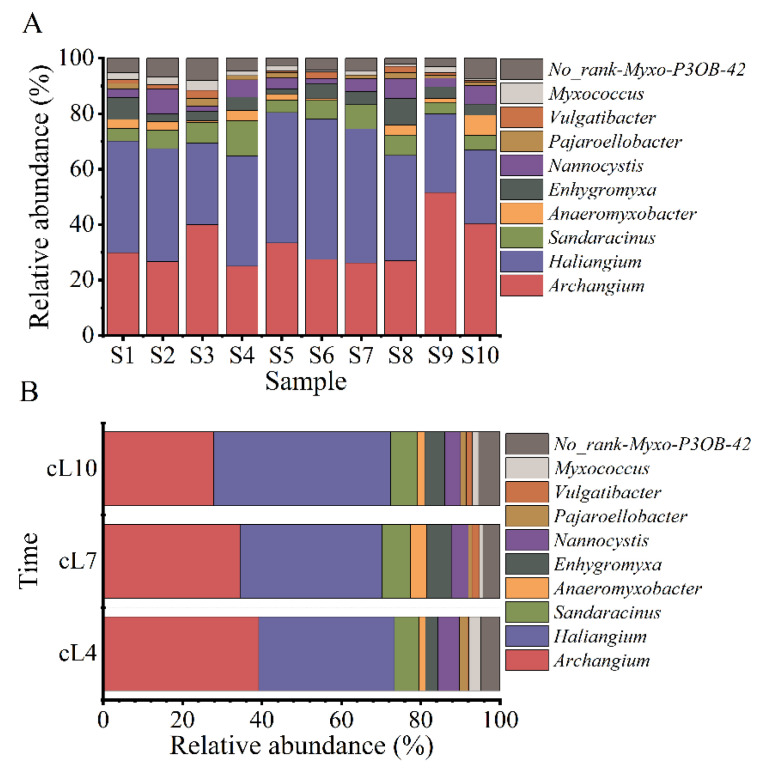
Temporal and spatial changes in the structure of the myxobacteria community. (**A**) The spatial distribution of the abundance of the top 10 genera of myxobacteria; (**B**) the temporal variation of the abundance of the top 10 genera of myxobacteria.

**Figure 3 microorganisms-11-01924-f003:**
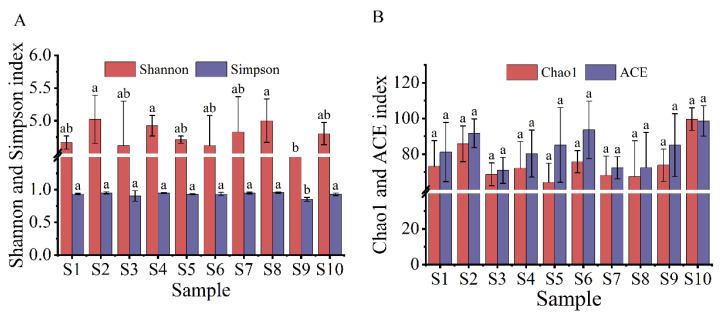
Spatial distribution of the α diversity of the myxobacteria community. (**A**) The spatial distribution of the Shannon index and Simpson index of the myxobacteria community; (**B**) the spatial distribution of the Chao1 index and ACE index in the community of myxobacteria. The lowercase letters (a,b) above the box represent significant differences (*p* < 0.05) of α diversity index of myxobacteria in different places.

**Figure 4 microorganisms-11-01924-f004:**
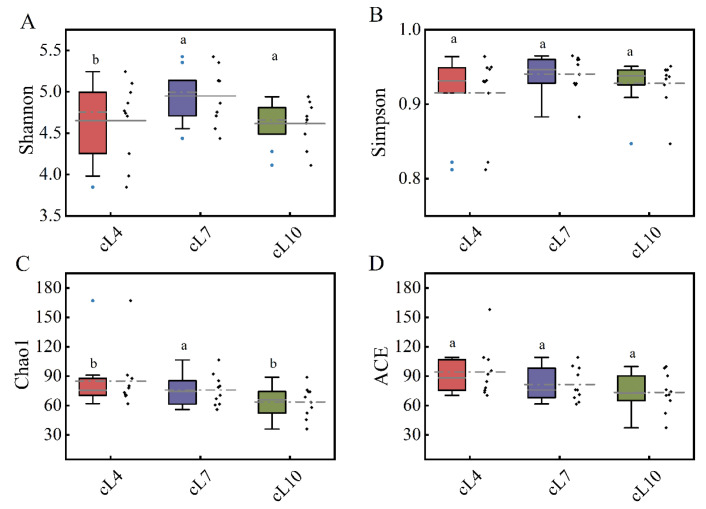
Temporal and spatial distribution of the α diversity of the myxobacteria community. (**A**) The Shannon index distribution of myxobacteria communities in April, July, and October; (**B**) the Simpson index distribution of the myxobacteria community in April, July, and October; (**C**) the Chao1 index distribution of the myxobacteria community in April, July, and October; (**D**) the ACE index distribution of the myxobacteria community in April, July, and October. The lowercase letters (a,b) above the box represent significant differences (*p* < 0.05) in the α diversity index of myxobacteria between months.

**Figure 5 microorganisms-11-01924-f005:**
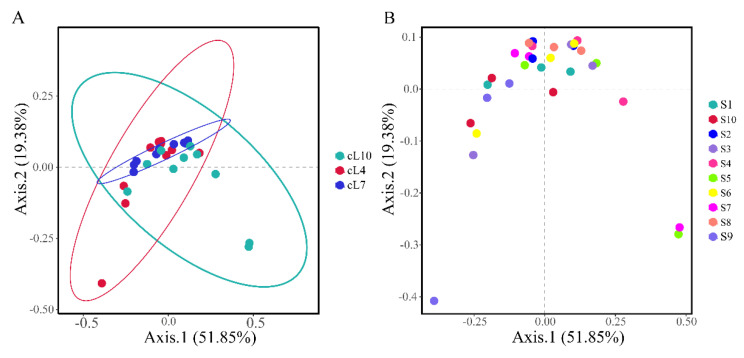
Principal component analysis (PCoA) of the spatiotemporal distribution of the myxobacteria community. (**A**) The temporal distribution of the Bray–Curtis myxobacteria community; (**B**) the spatial distribution of the Bray–Curtis myxobacteria community.

**Figure 6 microorganisms-11-01924-f006:**
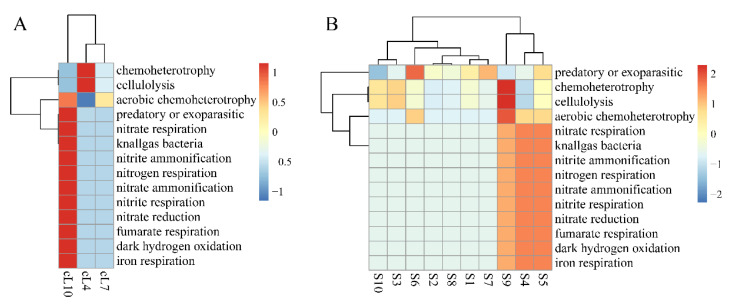
Spatial and temporal heat map of the FAPORTAX function of the Tamarix chinensis Lour rhizosphere in Ebinur Lake wetland. (**A**) The FAPORTAX functional distribution of myxobacteria in three months; (**B**) the functional distribution of myxobacteria in habitats S1–S10.

**Figure 7 microorganisms-11-01924-f007:**
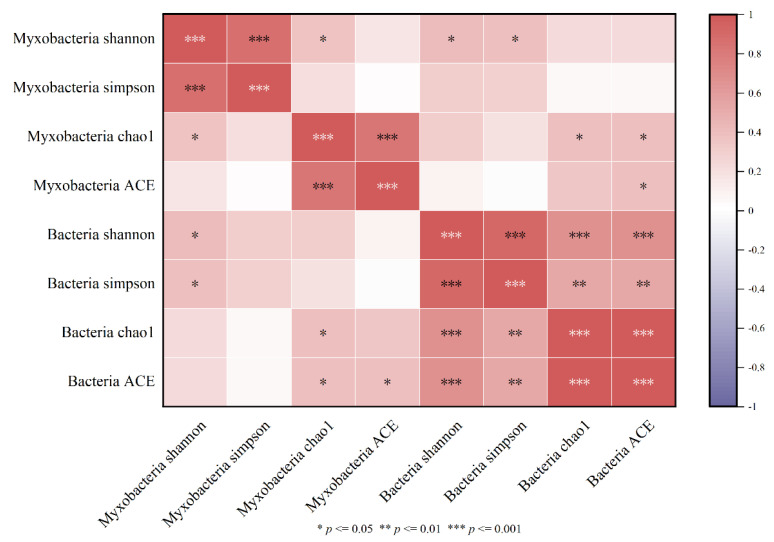
Heatmap of the myxobacterium diversity index and bacterial diversity correlation.

**Figure 8 microorganisms-11-01924-f008:**
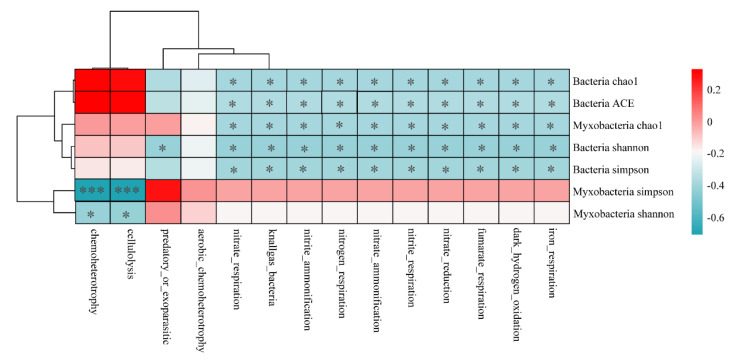
Heatmap of the functional abundance of myxobacteria associated with bacterial and myxobacterial diversity. * *p* < 0.05, *** *p* < 0.001.

**Figure 9 microorganisms-11-01924-f009:**
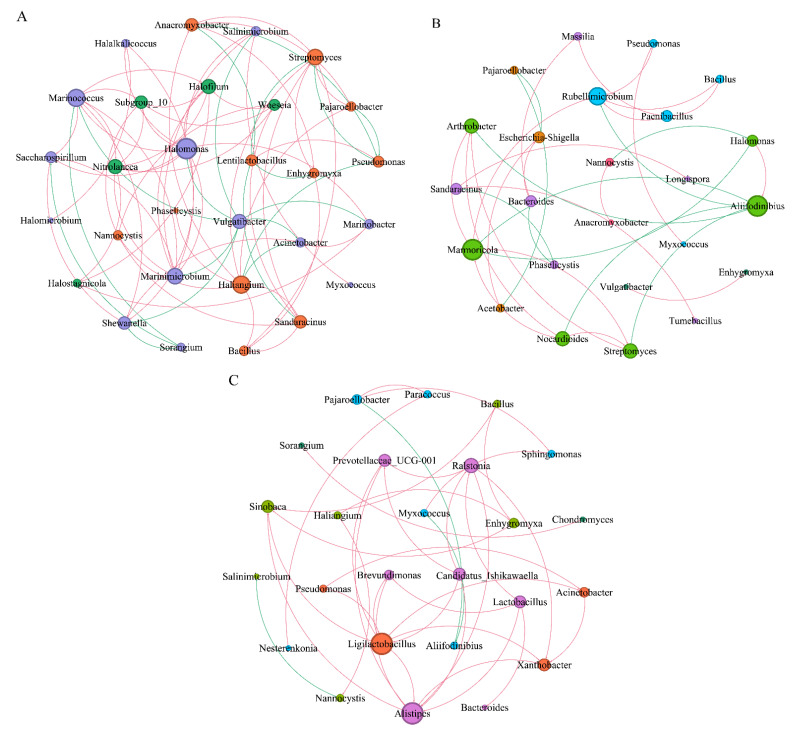
Interaction between myxobacteria and bacterial network in the rhizosphere of *Tamarix chinensis* Lour in Ebinur Lake wetland. (**A**) The symbiosis analysis of the myxobacteria genus and bacterial genus network in April; (**B**) the symbiosis analysis of the myxobacteria genus and bacterial genus network in July; (**C**) the symbiosis analysis of the Myxobacteria genus and bacterial genus network in October, with node size representing relative abundance, node coloring according to the modularity class process, edges representing the interaction between relative abundance, red lines indicating positive correlation, and green lines indicating negative correlation. The thickness of the edge represents the magnitude of the significant correlation coefficient between the relative abundance of two species.

**Figure 10 microorganisms-11-01924-f010:**
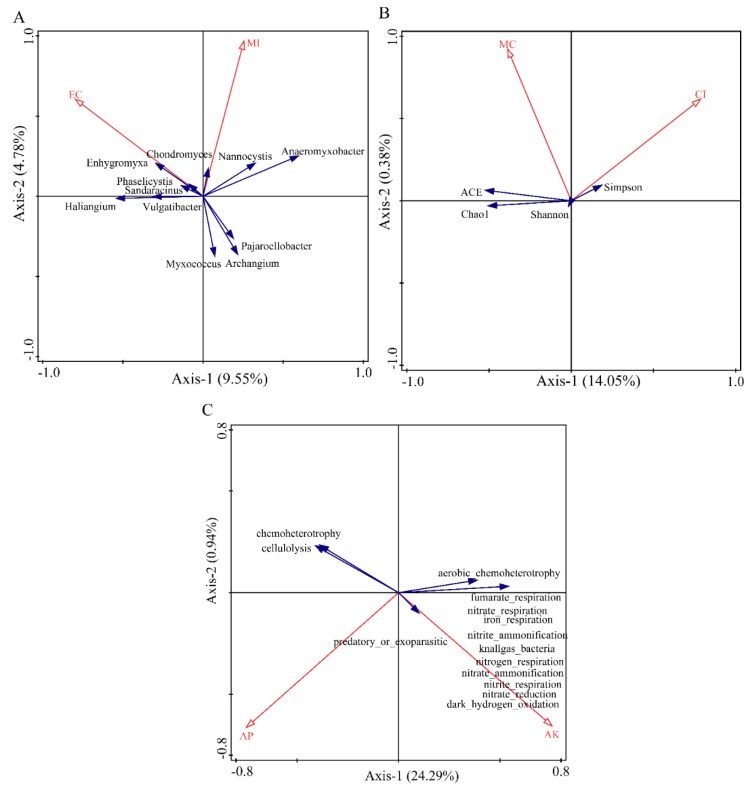
Redundancy analysis (RDA) of community structure, diversity, function, and soil physicochemical factors of myxobacteria in the rhizosphere of Tamarix chinensis Lour in Ebinur Lake wetland. (**A**) The redundancy analysis between the genus myxobacteria and soil physicochemical factors; (**B**) the redundancy analysis between myxobacteria diversity and soil physicochemical factors; (**C**) the redundancy analysis between myxobacteria function and soil physicochemical factors.

**Table 1 microorganisms-11-01924-t001:** Symbiosis parameters of the *Tamarix ramosissima* rhizosphere microbial network in Ebinur Lake wetland.

Season	Nodes	Edges	Degree	Network Distance	Density	Positive Correlation (%)	Negative Correlation (%)	Modularity
April	29	75	2.856	2.764	0.185	74.67%	25.34	0.419
July	24	32	1.333	2.524	0.116	65.63	34.38	0.574
October	25	37	1.480	2.572	0.123	91.89	8.10	0.515

## Data Availability

All data generated or analyzed during this study are included in this published article and its Appendix A.

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
