# Peer review of "Diversity and Functional Distribution Characteristics of Myxobacterial Communities in the Rhizosphere of Tamarix chinensis Lour in Ebinur Lake Wetland, China"

_microorganisms, 2023, doi:10.3390/microorganisms11081924_

Round 1
Reviewer 1 Report
Ref: microorganisms-2486991
This manuscript presents an interesting study of myxobacterial communities in the rhizosphere of Tamarix chinensis Lour in Lake Ebinur, focusing on their diversity and functional dispersal characteristics. The authors conducted a comprehensive analysis using molecular methods and functional profiling to reveal the diversity and potential functions of myxobacteria in this ecosystem. The study design is well structured and the methods used are appropriate to achieve the research objectives. The results are presented in a logical and concise manner and supported by relevant figures and tables. The discussion section provides valuable insights into the implications of the results and their significance in the context of myxobacterial ecology. However, there are some revisions that the authors should make before finalizing the manuscript:
Without line numbers, it is difficult for me to review minor points on specific lines in a page. However, there are major issues that should be clarified on priority. As follows:
Abstract:
1- While the abstract highlights how myxobacterial community structure and diversity can enhance the natural environment of wetlands, it would be helpful to directly highlight the existing knowledge gap or the importance of answering this research question.
2- Include a sentence about the methods used for high-throughput sequencing of the 16S rRNA gene.
3- The abstract mentions a number of results, such as the diversity of myxobacterial communities in different habitats and their functions, but it would be beneficial to elaborate on the main results.
4- Spell out the abbreviations EC, CI, and AK
Introduction:
1- The authors write, "Dai et al [8] analyzed myxobacterial community in six different fecal samples......" Please correct the cited paper where the myxobacterial population was analyzed in four different compost manures.
2- Please correct “Daiwei et al [10] on page 2.” I could not find this reference
3- Last paragaraph in page 2. Please correct “Therefore, the presents study aims….” to “Therefore, the present study aims to (1) study…(2) study…, and (3) study”
4- What practical applications or implications might the findings have for the management of the Ebinur Lake Wetland?
Materials and Methods:
Research location:
1- Explain the rationale for the five-point sampling method
Soil DNA extraction:
1- The authors stated “……by using an improved DNeasy®PoweeSoil®Pro kit [50]”. Please provide the correct reference. Your complete list consists of 29 cited references, how come this reference is 50?
2- Please provide the original reference for universal primers 515F and 907R (Beller et al. PMID: 23064329)
3- Please provide the number of PCR cycles
4- How did the authors manage to determine the concentrations of their target amplicons from the agarose gel?
5- What is IlluminaPE250? Do you mean “on a NovaSeq instrument (Illumina) at 2 × 250bp read length?”
Results:
1- Authors should provide information on the number of reads, chimeric sequences, valid reads, and read coverage
2- I do not see if the raw sequence data was rarefied and/or normalized? No robust results can be inferred if this has not been done
Discussion:
1- The authors mention the Shannon index as a measure of myxobacterial community diversity. They need to explain the ecological implications of the observed diversity and how it relates to the wetland ecosystem. In addition, a comparison of the diversity values obtained with other studies or a discussion of the factors that may contribute to the observed diversity would add depth to the discussion.
2- The authors briefly mention the functions predicted by the FAPROTAX analysis, but do not provide a detailed interpretation of these functions and their importance in the context of the wetland ecosystem. The authors need to explore the potential ecological role of myxobacteria in relation to the degradation of nitrogen compounds and the regulation of soil microorganisms
3- The authors note that there is a positive correlation between the α-diversity indices of myxobacteria and bacterial communities. They may need to explain why bacterial community diversity may influence myxobacterial community diversity. In addition, they also need to discuss the possible ecological mechanisms underlying this correlation.
4- The authors are requested to elaborate on the implications of the findings that show the co-occurrence of microbial networks involving Pseudomonas, Shewanella, and Halomonas interacting with myxobacteria. They also need to discuss the ecological roles of Proteobacteria as a main biological factor affecting Myxococcota
5- The authors mention the ability of myxobacteria to form myxospores and their resistance to harsh environmental conditions, which could mitigate the influence of environmental factors on myxobacteria. While this is a valid point, it would be helpful to discuss the limitations of the study and consider other potential factors or interactions (not measured in this study) that may also influence the myxobacterial community.
The English of the manuscript needs minor revision.
Author Response
Dear editor and reviewers
On behalf of my co-authors, we thank you very much for giving us an opportunity to revise our manuscript, and we also appreciate reviewers very much for their positive and constructive comments and suggestions on our manuscript entitled “Diversity and functional distribution characteristics of myxo-bacterial communities in the rhizosphere of Tamarix chinensis Lour in Ebinur Lake Wetland, China” (Manuscript Number: microorganisms-2486991). As you are concerned, there are several problems that need to be addressed. According to your nice suggestions, we have made extensive corrections to our previous draft, the detailed corrections are listed below. The reviewer comments are laid out below in italicized font and specific concerns have been numbered. To facilitate this discussion, our response is given in normal font and additions to the manuscript are given in the red text.
1.Abstract:
Point 1: While the abstract highlights how myxobacterial community structure and diversity can enhance the natural environment of wetlands, it would be helpful to directly highlight the existing knowledge gap or the importance of answering this research question.
Response 1: Thank you very much for your review. According to your comments, I have changed the abstract of the paper.
Point 2: Include a sentence about the methods used for high-throughput sequencing of the 16S rRNA gene.
Response 2: Thank you very much for your review. According to your comments, I have revised the manuscript in place accordingly (line 16-21 on page 1).
Point 3: The abstract mentions a number of results, such as the diversity of myxobacterial communities in different habitats and their functions, but it would be beneficial to elaborate on the main results.
Response 3: Thank you very much for your review. According to your comments, I have changed the abstract of the paper (line 21-26 on page 1).
Point 4: Spell out the abbreviations EC, CI, and AK
Response 4: Thank you for pointing out this problem in manuscript. Full writes of the EC, CI, and AK words are electrical conductivity, exchangeable calcium, available potassium (line 37 on page 3, line 31-34 on page 1).
2.Introduction:
Point 1: The authors write, "Dai et al [8] analyzed myxobacterial community in six different fecal samples......" Please correct the cited paper where the myxobacterial population was analyzed in four different compost manures.
Response 1: Thanks for your careful revision, I am sorry for my carelessness. According to your prompt, I have revised it accordingly in the manuscript (line 13-16 on page 2).
Point 2: Please correct “Daiwei et al [10] on page 2.” I could not find this reference
Response 2: According to the regulatory effect of Corallococcus sp. EGB on soil microorganisms, a literature that can be detected and matched has been replaced (line 21-23 on page 2, reference 10).
Point 3: Last paragaraph in page 2. Please correct “Therefore, the presents study aims….” to “Therefore, the present study aims to (1) study… (2) study…, and (3) study”
Response 3: Thanks for your careful checks. In your opinion, I have revised the manuscript in place accordingly (line 45-52 on page 2).
Point 4: What practical applications or implications might the findings have for the management of the Ebinur Lake Wetland?
Response 4: Thank you very much for your review. This study provides theoretical support for improving wetland salinization, improving soil microbial functions, and promoting the restoration of ecological vegetation in Ebinur Lake Wetland. The study lays the foundation for the future development and utilization of myxobacterial community.
3.Materials and Methods:
3.1Research location:
Point 1: Explain the rationale for the five-point sampling method
Response 1: Thank for your comments. The five-point sampling method means that the midpoint of the diagonal is determined as the central sampling point, and then four points equal to the distance of the diagonal are selected as the sample point. Near the three main inflow rivers of the Ebinur Lake Wetland (Bortala Mongol Autonomous Prefecture, Jinghe River and Kuitun River), 10 plots of 100 m 100 m were set up and marked as S1~S10, and Tamarix chinensis Lour soil samples were collected by five-point sampling method (line 5-7 on page 3).
3.2Soil DNA extraction:
Point 1: The authors stated “……by using an improved DNeasy®PoweeSoil®Pro kit [50]”. Please provide the correct reference. Your complete list consists of 29 cited references, how come this reference is 50?
Response 1: Thanks for your careful checks. I am sorry for my carelessness. Based on your comments, I have made the corrections to make the word harmonized within the whole manuscript (line 33 on page 3, reference 19).
Point 2: Please provide the original reference for universal primers 515F and 907R (Beller et al. PMID: 23064329)
Response 2: Thanks for your suggestion. In your opinion, I have replaced the original references for the universal primers 515F and 907R (line 34-36 on page 3, reference 20).
Point 3: Please provide the number of PCR cycles
Response 3: Thanks for your suggestion. According to your opinion, I have added the number of PCR cycles to the manuscript. The entire amplification process consisted of 30 PCR cycles (line 1-4 on page 4).
Point 4: How did the authors manage to determine the concentrations of their target amplicons from the agarose gel?
Response 4: Thank for your comments. The target amplicon was cut from an agarose gel plate, and the concentration of the dissolved target amplicon was determined by using an ultramicrospectrophotometer (NanoDrop2000C Spectrophometer).
Point 5: What is IlluminaPE250? Do you mean “on a NovaSeq instrument (Illumina) at 2 × 250bp read length?”
Response 5: I am sorry that this part was not clear in the original manuscript. I should have explained that the IlluminaPE250 platform is a high-throughput sequencing platform provided by Beijing Nuohe Zhiyuan. I have revised the contents of this part (line 11-12 on page 4).
4.Results:
Point 1: Authors should provide information on the number of reads, chimeric sequences, valid reads, and read coverage
Response 1: High-throughput sequencing data have been uploaded to the NCBI database (BioProject: PRJNA997573). SampleSeq_info is the corresponding barcode and primer sequences of each sample, and QCstat.xls is the data pre-processing statistics and quality control information In the Supplementary files. Special note on the coverage of myxobacteria (see alpha_diversity_index document for data), we selected from the bacterial sequence. since there is no universal primer for all myxobacteria, the bacterial coverage has met the requirements and the bacterial Shannon curve is flattened, indicating that the amount and depth of sequencing data can reflect the real situation of soil myxobacteria.
Point 2: I do not see if the raw sequence data was rarefied and/or normalized? No robust results can be inferred if this has not been done
Response 2: The raw data obtained by sequencing has a certain proportion of interference data. In order to make the results of information analysis more accurate and reliable, the raw data is first spliced and filtered to obtain the effective data (Clean Data), so all the raw data are normalized. Pre-processing of sequencing data mainly includes splicing and quality control of the data (Raw PE) obtained by Illumina NovaSeq sequencing, obtaining Clean Tags, and then performing chimera filtering to obtain effective data that can be used for subsequent analysis (Effective Tags). Data preprocessing statistics and quality control results are detailed in the supplementary files.
5.Discussion:
Point 1: The authors mention the Shannon index as a measure of myxobacterial community diversity. They need to explain the ecological implications of the observed diversity and how it relates to the wetland ecosystem. In addition, a comparison of the diversity values obtained with other studies or a discussion of the factors that may contribute to the observed diversity would add depth to the discussion.
Response 1: I sincerely appreciate the valuable comments. Based on your comments, I explained the ecological implications of the observed diversity and how it relates to the wetland ecosystem, I supplemented the factors that may contribute to the observed diversity (line 13-22 on page 14, reference 32).
Point 2: The authors briefly mention the functions predicted by the FAPROTAX analysis, but do not provide a detailed interpretation of these functions and their importance in the context of the wetland ecosystem. The authors need to explore the potential ecological role of myxobacteria in relation to the degradation of nitrogen compounds and the regulation of soil microorganisms
Response 2: Thank you for the above suggestions. We explained these functions and their importance in the wetland ecosystems. Since the ecological function of myxobacteria is rarely studied, we initially explored the potential ecological function of myxobacteria based on FAPROTAX function (line 40-42 on page 14, Line 21-25 on page 15).
Point 3: The authors note that there is a positive correlation between the α-diversity indices of myxobacteria and bacterial communities. They may need to explain why bacterial community diversity may influence myxobacterial community diversity. In addition, they also need to discuss the possible ecological mechanisms underlying this correlation.
Response 3: Thank you very much for your comments, and with your guidance, I explained why bacterial community diversity can affect myxobacteria community diversity and discuss possible potential ecological mechanisms (line 5-14 on page 15, reference 8, 13, 34,35).
Point 4: The authors are requested to elaborate on the implications of the findings that show the co-occurrence of microbial networks involving Pseudomonas, Shewanella, and Halomonas interacting with myxobacteria. They also need to discuss the ecological roles of Proteobacteria as a main biological factor affecting Myxococcota.
Response 4: Thank you for the above suggestions. In your opinion, I supplemented the symbiosis of Pseudomonas, Shewanella, and Halomonas with the myxacterial network, and reviewed the literature to supplement the main ecological role of Proteobacteria (line 17-25 on page 15, reference7,31,33).
Point 5: The authors mention the ability of myxobacteria to form myxospores and their resistance to harsh environmental conditions, which could mitigate the influence of environmental factors on myxobacteria. While this is a valid point, it would be helpful to discuss the limitations of the study and consider other potential factors or interactions (not measured in this study) that may also influence the myxobacterial community.
Response 5: As suggested by the reviewer, we have added more references to support this idea (line 49-51 on page 15, line 1-6 on page 15references 11,31,37).
Comments on the Quality of English Language:
Response: The English of the manuscript needs minor revision.
Response: Thanks for your suggestion. We do invite a friend of us who is a native English speaker (MJEditor (www.mjeditor.com))from the USA to help polish our article. And we hope the revised manuscript could be acceptable for you.
We tried our best to improve the manuscript and made some changes marked in red in revised paper which will not influence the content and framework of the paper. We appreciate for Editors and Reviewers’ warm work earnestly, and hope the correction will meet with approval. Once again, thank you very much for your comments and suggestions.
Sincerely,
Xuemei Chen, first author
Corresponding author. Wenge Hu
E-mail address: wengehushiheziu@163.com
School of Life Science, Shihezi University, Shihezi, Xinjiang, China, 832000

Reviewer 2 Report
The authors have reported about impact of microbes in Ebinur Lake Wetland through general metagenomic analysis in different time point, in this manuscript titled with " Diversity and functional distribution characteristics of myxobacterial communities in the rhizosphere of Tamarix chinensis Lour in Ebinur Lake Wetland, China". The authors have done comprehensive bioinformatic analysis by following the QIIME protocol and successfully associated the relation between microbes and Ebinur Lake Wetland.
Authors have followed QIIME protocol in detail and provided all analysis images but explanation in detail into the result san discussions are missing. For example, RDA can provide insights into the relationships between environmental factors and microbial communities, allowing researchers to identify which variables are driving the observed community composition. It helps in understanding the ecological factors shaping microbial diversity, community structure, and their response to environmental changes. Apart from providing the figure of RDA, author should write into the manuscript in details about finding in figure.
Authors should keep some of the figures is supplementary files. For example figure 4, Principal component analysis (PCoA) of the spatiotemporal distribution of the myxobacteria community does not showing significant difference it should be in supplementary file.
Author should also provide the all supplementary files excel sheets.
English writing part of the manuscript is good.
Author Response
Dear editor and reviewers
On behalf of my co-authors, we thank you very much for giving us an opportunity to revise our manuscript, and we also appreciate reviewers very much for their positive and constructive comments and suggestions on our manuscript entitled “Diversity and functional distribution characteristics of myxo-bacterial communities in the rhizosphere of Tamarix chinensis Lour in Ebinur Lake Wetland, China” (Manuscript Number: microorganisms-2486991). As you are concerned, there are several problems that need to be addressed. According to your nice suggestions, we have made extensive corrections to our previous draft, the detailed corrections are listed below. The reviewer comments are laid out below in italicized font and specific concerns have been numbered. To facilitate this discussion, our response is given in normal font and additions to the manuscript are given in the red text.
Point 1: Authors have followed QIIME protocol in detail and provided all analysis images but explanation in detail into the result san discussions are missing. For example, RDA can provide insights into the relationships between environmental factors and microbial communities, allowing researchers to identify which variables are driving the observed community composition. It helps in understanding the ecological factors shaping microbial diversity, community structure, and their response to environmental changes. Apart from providing the figure of RDA, author should write into the manuscript in details about finding in figure.
Response 1. We are grateful for the suggestion. To be more clearly and in accordance with the reviewer concerns, we have added a more detailed interpretation to the manuscript. More detailed statistical analysis was added on page (Line 13-22, page 14; Line 40-43, page 14; Line 5-14, Line 17-25, Line 48-52, page 15; Line 1-6, page 16).
Point 2: Authors should keep some of the figures is supplementary files. For example, figure 4, Principal component analysis (PCoA) of the spatiotemporal distribution of the myxobacteria community does not showing significant difference it should be in supplementary file.
Response 2: Thanks for your suggestion. Principal component analysis (PCoA) of the spatiotemporal distribution of the myxobacteria community are detailed in Table S1 and Table S2 in the Supplementary file (Line 8-12 on page 7).
Point 3: Author should also provide the all supplementary files excel sheets.
Response 3: We sincerely appreciate the valuable comments. In your opinion, we have put all the supplementary material into the folder "supplymentary files".
Point 4: English writing part of the manuscript is good.
Response 4: We sincerely appreciate the valuable comments. The language presentation was improved with assistance from a native English speaker with appropriate research background (MJEditor (www.mjeditor.com)).
We tried our best to improve the manuscript and made some changes marked in red in revised paper which will not influence the content and framework of the paper. We appreciate for Editors and Reviewers’ warm work earnestly, and hope the correction will meet with approval. Once again, thank you very much for your comments and suggestions.
Sincerely,
Xuemei Chen, first author
Corresponding author. Wenge Hu
E-mail address: wengehushiheziu@163.com
School of Life Science, Shihezi University, Shihezi, Xinjiang, China, 832000

Reviewer 3 Report
Please respond to the following remarks;
- Abbreviate the abstract
- Movie the sentences relevant to the study area from the Introduction section to the first of the materials and methods. I recommend addition a location map showing the investigated lake and sampling sites.
- Reduce the subtitles as possible.
_ Based on the study main findings specifically;h
- It was mentioned in the conclusion that "It has been found in this study that the microbial community function in rhizosphere soils is mainly related to the nitrogen cycle and myxobacteria plays ani mportant role in promoting the soil nitrogen cycle. EC, CI, and AK were determined as the main abiotic factors affecting the diversity, structure, and function of myxobacterial communities". These findings need recommendations.
Author Response
Dear editor and reviewers
On behalf of my co-authors, we thank you very much for giving us an opportunity to revise our manuscript, and we also appreciate reviewers very much for their positive and constructive comments and suggestions on our manuscript entitled “Diversity and functional distribution characteristics of myxo-bacterial communities in the rhizosphere of Tamarix chinensis Lour in Ebinur Lake Wetland, China” (Manuscript Number: microorganisms-2486991). As you are concerned, there are several problems that need to be addressed. According to your nice suggestions, we have made extensive corrections to our previous draft, the detailed corrections are listed below. The reviewer comments are laid out below in italicized font and specific concerns have been numbered. To facilitate this discussion, our response is given in normal font and additions to the manuscript are given in the red text.
Point 1: Abbreviate the abstract
Response 1: Thank you for making this valuable suggestion. In your opinion, we have abbreviated the paper abstract.
Point 2: Movie the sentences relevant to the study area from the Introduction section to the first of the materials and methods. I recommend addition a location map showing the investigated lake and sampling sites.
Response 2: Thank you for this very insightful comment. According to your request, we use GS1.0 software to map the Lake wetlands and sampling points.
Point 3: Reduce the subtitles as possible.
Response 3: I sincerely appreciate the valuable comments. We have further revised the manuscript in your opinion.
Point 4: Based on the study main findings specifically
Response 4: Thank you for this very insightful comment. We further revised the manuscript based on the main study findings.
Point 5: It was mentioned in the conclusion that "It has been found in this study that the microbial community function in rhizosphere soils is mainly related to the nitrogen cycle and myxobacteria plays an important role in promoting the soil nitrogen cycle. EC, CI, and AK were determined as the main abiotic factors affecting the diversity, structure, and function of myxobacterial communities". These findings need recommendations.
Response 5: Thank you very much for your correction, based on your opinion, we have added references to support our argument. the microbial community function in rhizosphere soils is mainly related to the nitrogen cycle (Line 40-51, page 14; Line 17-25, page 15); myxobacteria plays an important role in promoting the soil nitrogen cycle. EC, CI, and AK were determined as the main abiotic factors affecting the diversity, structure, and function of myxobacterial communities (Line 35-45, reference7,8,31,29,37).
We tried our best to improve the manuscript and made some changes marked in red in revised paper which will not influence the content and framework of the paper. We appreciate for Editors and Reviewers’ warm work earnestly, and hope the correction will meet with approval. Once again, thank you very much for your comments and suggestions.
真诚地
陈雪梅,第一作者
通讯作者。胡文戈
电子邮件地址: wengehushiheziu@163.com
石河子大学生命科学学院, 新疆石河子, 832000
